# Sizes, Components, Crystalline Structure, and Thermal Properties of Starches from Sweet Potato Varieties Originating from Different Countries

**DOI:** 10.3390/molecules27061905

**Published:** 2022-03-15

**Authors:** Yibo Li, Lingxiao Zhao, Laiquan Shi, Lingshang Lin, Qinghe Cao, Cunxu Wei

**Affiliations:** 1Key Laboratory of Crop Genetics and Physiology of Jiangsu Province/Joint International Research Laboratory of Agriculture & Agri-Product Safety of the Ministry of Education, Yangzhou University, Yangzhou 225009, China; dx120180133@yzu.edu.cn (Y.L.); mz120201523@yzu.edu.cn (L.S.); 007520@yzu.edu.cn (L.L.); 2Co-Innovation Center for Modern Production Technology of Grain Crops of Jiangsu Province/Jiangsu Key Laboratory of Crop Genomics and Molecular Breeding, Yangzhou University, Yangzhou 225009, China; 3Xuzhou Institute of Agricultural Sciences in Jiangsu Xuhuai District, Xuzhou 221131, China; zhaolxiao2019@163.com

**Keywords:** sweet potato, germplasm, starch, crystalline structure, thermal properties

## Abstract

Sweet potato is a root tuber crop and an important starch source. There are hundreds of sweet potato varieties planted widely in the world. Starches from varieties with different genotype types and originating from different countries have not been compared for their physicochemical properties. In the research, starches from 44 sweet potato varieties originating from 15 countries but planted in the same growing conditions were investigated for their physicochemical properties to reveal the similarities and differences in varieties. The results showed that the 44 starches had granule size (D[4,3]) from 8.01 to 15.30 μm. Starches had different iodine absorption properties with OD680 from 0.259 to 0.382 and OD620/550 from 1.142 to 1.237. The 44 starches had apparent amylose content from 19.2% to 29.2% and true amylose content from 14.2% to 20.2%. The starches exhibited A-, C_A_-, C_C_-, or C_B_-type X-ray diffraction patterns. The thermograms of 44 starches exhibited one-, two-, or three-peak curves, leading to a significantly different gelatinization temperature range from 13.1 to 29.2 °C. The significantly different starch properties divide the 44 sweet potato varieties into different groups due to their different genotype backgrounds. The research offers references for the utilization of sweet potato germplasm.

## 1. Introduction

Sweet potato (*Ipomoea batatas*), an important root tuber crop, provides food and energy for people, especially in Asia and Africa [1]. Sweet potato contains 15–30% and 46–68% starch in its wet and dry root tuber, respectively, among different varieties, and has become the first choice for producing starch due to its short growth cycle, strong environment adaptability, low planting cost, and high yield in the world, especially in developing countries [2,3]. Starch isolated from the sweet potato root tuber has been used to produce noodles and vermicelli. In addition, it is also a good thickening agent in cooking foods and an important raw material in producing syrup, film, lactic acid, ethanol, and other chemicals [4,5]. The applications of starch are influenced by its physicochemical properties including size, components, crystalline structure, and thermal properties [3,6,7,8,9].

Sweet potato is planted widely in the world, and has hundreds of varieties or lines due to artificial selection, natural hybrids, and mutations [1,4]. The root tuber of sweet potato has different colored skin and flesh, but starch properties have no significant correlation with skin and flesh color, and are determined by the genotypes of varieties [2,10,11]. Starches from different sweet potato varieties have been studied for their structures, functional properties, and applications [10,12,13,14]. For example, Collado et al. [12] investigated the genetic variation in physical properties of sweet potato starches from 44 genotypes adapted to Philippine conditions, and found that wide variation and distinctly different pasting properties exist among genotypes. Zhu and Xie [14] studied the swelling power, water solubility, rheological properties, gelatinization, and retrogradation of starches from seven New Zealand sweet potatoes, and concluded that the starch properties from different varieties exhibit significant diversity due to their different internal unit chain parameters of amylopectin. Kim et al. [13] compared the physicochemical properties of starches from eight Korean sweet potato varieties including purple-, orange-, and white/cream-fleshed tubers. The starches exhibit polygonal and semi-oval shapes with different granule sizes among different varieties, and have A- and C_B_-type X-ray diffraction patterns. The pasting properties, amylose contents and water binding capacities of starches are closely related to the chain length distribution of amylopectin, and have no relationship with tuber color. Similar results are also reported in starches from eight Chinese sweet potato varieties with light yellow-, orange-, and purple-fleshed tubers. The molecular structure of amylose and amylopectin exhibits differences among different varieties, and is the main influencing factor in determining starch physicochemical properties [10]. Though the above references have reported the differences in starches from different varieties, the varieties originate from the same country or region in the same report. At present, no research reports the starch properties of sweet potato varieties originating from different countries. The growing environment and field management significantly influence the development and properties of starch in plant resources [15,16,17,18]. In addition, the measuring and analysis methods of starch properties also affect the property parameters. Therefore, it is necessary to study the characteristics of starches from sweet potato varieties originating from different countries but planted in the same growing conditions.

In this research, 44 sweet potato varieties originating from 15 countries were planted in the same conditions, and their starches were isolated and investigated for granule size, iodine absorption, amylose content, crystalline structure, and thermal properties. The hierarchical cluster analysis was carried out to reveal the differences in sweet potato varieties based on the starch property parameters. The objective of this study was to evaluate the germplasm resources of sweet potato starch. This research offers some references for the utilization of sweet potato germplasm.

## 2. Results and Discussion

### 2.1. Granule Morphology and Size Distribution of Starch

The granule morphology and size are important properties of starch, affecting functional properties and applications of starch [9]. The morphologies of isolated starches were observed under normal and polarized light (Figure 1). Starch granules had round, polygonal, oval, and semi-oval shapes, and contained small and large granules with typical “Maltese crosses” having the hila in the center of the granules. No significant differences in starch morphology were observed among all sweet potato varieties. Similar morphology has also been reported in sweet potato starch [19,20]. However, the granule size distributions exhibited significant differences among some sweet potato starches (Table 1). The surface- (D[3,2]) and volume-weighted mean diameter (D[4,3]) are usually used to indicate the size of starch, and ranged from 3.866 to 7.681 μm and from 8.013 to 15.296 μm among 44 sweet potato starches, respectively. The starch size in this study agreed with the previous report of sweet potato starch [2,19,20]. The starch size is influenced by plant source, variety genotype, plant physiology, and growing environment [9,21,22].

### 2.2. Iodine Absorption and Amylose Content of Starch

Starch contains amylose and amylopectin. Both amylose and amylopectin influence the absorption of starch and iodine [23]. In this research, the iodine absorption spectra of 44 starches were analyzed (data not shown), and the iodine absorption parameters including the value of optical density (OD) at 550 nm (OD550), 620 nm (OD620), and 680 nm (OD680) were measured. The OD680 is usually defined as blue value (BV) of starch, and displays the iodine binding capacity of starch [24]. The ratio of OD620 to OD550 (OD620/550) can reflect the relative proportion of long chains in starch [23]. The OD680 and OD620/550 ranged from 0.259 to 0.382 and from 1.142 to 1.237 among 44 starches, respectively (Table 2), indicating that amylose and amylopectin were different among these starches.

Amylose content is the important structure parameter, determining the applications of starch [25,26]. The amylose content is usually determined with the iodine colorimetry method according to the OD620 of starch–iodine complex. However, the iodine colorimetry usually overestimates the amylose content of starch because of the branch-chains of amylopectin that can bind the iodine. Therefore, the amylose content determined by iodine colorimetry is called the apparent amylose content (AAC) [23]. In this study, the AAC varied from 19.2% to 29.2% (Table 2), and is in the range of sweet potatoes reported by previous references [14,19,27]. The concanavalin A (Con A) precipitate method can accurately determine the amylose content to avoid the overestimation of amylose content. Using this method, starch is completely dissolved into amylose and amylopectin, and separated into two aliquots. The Con A specifically binds and precipitates the amylopectin in an aliquot, and the remaining amylose in the supernatant is hydrolyzed by amylolytic enzymes to glucose, which is measured with glucose oxidase/peroxidase reagent for determining the quality of amylose. The amylose and amylopectin in another aliquot are similarly hydrolyzed to glucose for measuring their quality. The amylose content is the quality percentage of amylose to both amylose and amylopectin. The measured amylose content is not influenced by the purity and moisture of starch, and is usually defined as true amylose content (TAC) [23]. The TAC ranged from 14.2% to 20.2% among 44 sweet potato starches (Table 2). The TAC was significantly lower than AAC in all starches, and the ΔAC (AAC–TAC) was significantly different among different varieties from 4.0% to 11.8%, indicating that varieties with different genotype backgrounds had significantly different amylopectin structure with different iodine binding capacity. The amylose is biosynthesized simply by granule-bound starch synthase I (GBSSI) encoded by the *Waxy* (*Wx*) gene in plant storage tissue, and amylopectin is biosynthesized complexly by soluble starch synthases (SSSs), starch branching enzymes (SBEs), and starch debranching enzymes (DBEs) [28]. There are many *Wx* alleles responsible for amylose synthesis from low to high in different varieties of the same species [29]. The SSSs, SBEs, and DBEs have many isoforms, and their quality and activities determine the amylopectin structure [30]. In this research, sweet potato varieties originate from different countries and areas. Their different genotype backgrounds with different qualities and activities of GBSSI, SSSs, SBEs, and DBEs led to different amylose contents and amylopectin structures among 44 sweet potato varieties.

The hierarchical cluster analysis based on OD680, OD620/550, AAC, and TAC was carried out to investigate the differences and similarities between sweet potato varieties (Figure 2). The 44 sweet potato varieties were divided into cluster 1 (C1) and cluster 2 (C2). The C2 contained 16 sweet potato varieties with OD680 from 0.259 to 0.319, OD620/550 from 1.167 to 1.237, AAC from 19.2% to 25.3%, and TAC from 15.0% to 20.2%. The C1 was further divided into two subgroups of C1A and C1B. The C1A contained eight sweet potato varieties with OD680 from 0.303 to 0.382, OD620/550 from 1.146 to 1.186, AAC from 25.0% to 29.2%, and TAC from 14.2% to 17.7%, and the C1B contained 20 varieties with OD680 from 0.284 to 0.326, OD620/550 from 1.142 to 1.222, AAC from 21.8% to 28.3%, and TAC from 14.5% to 19.7% (Figure 2; Table 2). The ΔAC ranged from 4.0% (PE04) to 6.6% (US02) in C2, from 6.8% (US06) to 9.6% (NG01) in C1B, and from 10.1% (PH01) to 11.8% (JP06) in C1A (Table 2), indicating that amylopectin structure plays an important role in influencing starch components and iodine absorption. Though some varieties originated from the same area, their genotypes exhibited significant differences. For example, for 15 varieties (JP01-JP15) originating from Japan, there were three, six, and six varieties in C1A, C1B and C2, respectively.

### 2.3. Crystalline Structure of Starch

The branch-chains of amylopectin can form A- and B-type crystallinity, which can be detected by X-ray diffraction (XRD). The XRD patterns of 44 starches are presented in Figure 3. Starches from plant tissues have A-, B-, and C-type. A- and B-type starch contains only A- and B-type crystallinity, respectively, but C-type starch simultaneously contains both A- and B-type crystallinities. According to the ratio of A- to B-type crystallinities from high to low, C-type starch is usually further classified into C_A_-, C_C_-, and C_B_-type. The C_C_-type starch has diffraction peaks at 2θ 5.6°, 15°, 17°, and 23°; C_A_-type starch exhibits significant shoulder peak at 2θ 18°, a characteristic peak of A-type crystallinity; and C_B_-type starch has significant shoulder peaks at 2θ 22° and 24°, two characteristic peaks of B-type crystallinity [7,31]. In this study, 44 sweet potato varieties had A-, C_A_-, C_C_-, and C_B_-type starches (Figure 3). The JP14 starch was A-type, the PE06 starch was C_B_-type, the AO01, CB01, CN02, JP01, JP06, JP12, JP13, JP15, ML01, PE01, PE04, PH01, PH02, TH01, TZ01, US03, US06, and US07 starches were C_A_-type, and the other starches were C_C_-type (Figure 3). The A-, C_A_-, C_C_-, and C_B_-type starches have been reported in sweet potato varieties [10,13,14,19], but the B-type starch has not been reported in references. Genkina et al. [16,32] reported that the soil temperature determines the crystallinity of sweet potato starch. The root tuber synthesizes A-type starch in soil temperature above 33 °C and B-type starch in soil temperature below 15 °C. In fact, the temperature, moisture, and amylopectin branch-chain length all influence the crystal conformation during starch synthesis. The low temperature, wet condition, and amylopectin long branch-chains tend to form B-type crystallinity, and the high temperature, dry condition, and amylopectin short branch-chains tend to form A-type crystallinity [33]. In this study, all varieties grew in the same environment, and underwent the same soil temperature and moisture variation during starch synthesized and accumulated. The expression and activities of starch synthesis related enzymes were different among the 44 sweet potato varieties because their different genotypes responded to the variation in soil temperature differently, leading to different crystalline structures among the 44 sweet potato starches. The RC of starch was measured, and ranged from 15.9% to 27.3% among 44 starches (Figure 3). The RC of starch is affected by amylose, amylopectin, crystallinity, and granule size [7,24,34].

### 2.4. Thermal Properties of Starch

The thermal properties are mainly functional properties and an important index for evaluating starch applications. The differential scanning calorimetric (DSC) thermograms of the 44 starches are shown in Figure 4. Significantly different thermograms were observed among the 44 sweet potato starches. These thermograms were divided into three types of one-peak thermogram (namely, JP14, JP06, ML01, and MY01), two-peak thermogram (namely, JP04, CN03, US04, and PE05), and three-peak thermogram (namely, JP10, AO01, US01, and TZ01) according to their peak patterns (Figure 4). The gelatinization onset (To), peak (Tp), and conclusion temperature (Tc) ranged from 52.2 to 73.8 °C, from 59.7 to 80.0 °C, and from 75.3 to 86.9 °C among the 44 varieties, respectively. The gelatinization temperature range (ΔT) varied from 13.1 to 29.2 °C, and the gelatinization enthalpy (ΔH) ranged from 10.7 to 16.9 J/g (Table 3). The large gelatinization parameter variation among the 44 varieties was mainly due to the significantly different thermal curves. Similar thermal properties have been shown in different sweet potatoes from different references [13,19,35,36,37]. For example, Osundahunsi et al. [37] reported one-peak thermogram with gelatinization temperature from 67 to 75 °C in sweet potato varieties white-skinned TIS-1499 and red-skinned TIB-2. Kim et al. [13] reported the ΔT from 13.0 to 36.7 °C among eight sweet potato varieties. Lee and Lee [36] reported the ΔT from 24.0 to 28.0 °C among three sweet potato varieties. Duan et al. [35] reported a two-peak DSC curve in sweet potato variety Jishu 25. Genkina et al. [32] fitted the two-peak DSC curve of sweet potato starch into two gelatinization peaks with the low gelatinization temperature peak for B-type crystallinity and the high gelatinization temperature peak for A-type crystallinity. Guo et al. [38] fitted one-, two-, and three-peak DSC curves of different sweet potato starches into three gelatinization peaks, and the low, middle, and high gelatinization temperature peaks are the gelatinization of B-, C-, and A-type starch granules, respectively, meaning that sweet potato root tuber contains A-, B-, and C-type starch. In the present study, the different proportions of A-, B-, and C-type starch in sweet potatoes led to the different patterns of DSC curves among the 44 sweet potato starches.

The hierarchical cluster analysis based on Tp, Tc, ΔT, and ΔH was carried out to investigate the differences and similarities between sweet potato varieties (Figure 5). The JP14 was divided into one cluster (C2), and exhibited significant differences to the other sweet potato varieties in cluster 1 (C1) because JP14 starch had a typical one-peak DSC pattern with the highest gelatinization temperature and the most narrow ΔT among the 44 sweet potato starches. The cluster 1 contained two subclusters of C1A and C1B. The C1A had 21 sweet potato varieties with Tp, Tc, ΔT, and ΔH from 59.7 to 77.1 °C, from 75.3 to 86.7 °C, from 22.0 to 29.2 °C, and from 10.7 to 15.0 J/g, respectively. The C1B had 22 sweet potato varieties with Tp, Tc, ΔT, and ΔH from 66.8 to 77.5 °C, from 78.8 to 85.4 °C, from 18.0 to 23.9 °C, and from 13.2 to 16.9 J/g, respectively. It is noteworthy that some varieties from the same country, such as the JP01-JP15 varieties originating from Japan, were distributed into different groups, indicating that varieties from the same country had significantly different thermal properties.

### 2.5. Relationships of Starch Properties and Cluster Analysis of Sweet Potato Varieties

The relationships between starch properties were analyzed (Table 4). Among the 44 sweet potato starches, the granule size (D[4,3]) had no significant relationship with amylose content, RC, and thermal properties except the ΔT. The OD680 was significantly positively correlated to AAC, TAC, and ΔH, the OD620/550 was negatively correlated to AAC, TAC, Tp, Tc, and ΔH, and the AAC, TAC, and RC were positively correlated to Tp, Tc, and ΔH. Amylose and amylopectin are important components in starch, especially amylose, its content significantly affects the physicochemical properties and applications of starch [23,25,26]. Although granule size also influences the starch thermal properties [9,34], the thermal properties of starch were mainly affected by amylose content (AAC, TAC), amylopectin structure (OD620/550), and crystal structure (RC) in sweet potato (Table 4).

The hierarchical cluster analysis based on D[4,3], OD680, OD620/550, AAC, TAC, RC, Tp, Tc, ΔT, and ΔH was further carried out to investigate the differences and similarities between sweet potato varieties (Figure 6). The JP14 was divided into cluster 2 (C2), and the other sweet potato varieties were divided into cluster 1 (C1), indicating that JP14 had significantly different starch properties, especially for its thermal properties. The result also agreed with their crystalline structure that only JP14 had A-type starch. In cluster 2, the 43 sweet potato varieties were divided into different groups, showing that they had different starch properties due to different genotype backgrounds.

Though the 44 sweet potato varieties were divided into different clusters according to their starch properties (Figure 6), it is noteworthy that the growing conditions significantly affect starch properties in sweet potato, especially for temperature [16] and fertilizer treatment [21]. Recent study also shows that starch physicochemical properties are affected significantly by varieties, growing locations, and their interaction in sweet potato [39]. In this study, the sweet potato varieties originating from different countries were planted in the same place and cultivated conditions to avoid the effects of growing environment on starch properties. However, the selected growing conditions are significantly different from growth habits of varieties in their original country. Therefore, the starch properties of sweet potato varieties in the present study might be different from those in their original countries.

## 3. Materials and Methods

### 3.1. Plant Materials

The 44 sweet potato varieties were randomly chosen, and their fresh root tubers were provided by Sweetpotato Research Institute, China Agricultural Academy of Sciences. These germplasm resources are all public varieties or landraces, and conserved in National Sweetpotato Genebank in Xuzhou, China. Among the 44 varieties, 15 were from Japan, 7 from United States, 6 from Peru, 4 from China, 2 Philippines, and the others from Angola, Argentina, Cambodia, Congo, Morocco, Mali, Malaysia, Nigeria, Thailand, and Tanzania. Their accession IDs and original areas are listed in Table 5. These sweet potato varieties were planted simultaneously in the farm of Xuzhou Sweetpotato Research Center (32°27′ N, 117°29′ E), Jiangsu Province, China, on April 28, and harvested on 26 October 2020. The soil in the experimental field was yellow fluvo-aquic soil with a sandy texture. The available K, available P, hydrolysable N, total N, and organic matter in 0–20 cm soil layer before the experiment were 95 mg kg^−1^, 19.7 mg kg^−1^, 94.2 mg kg^−1^, 1.06 g kg^−1^, and 16.2 g kg^−1^, respectively, and its pH was 7.54. The fertilizer including 75 kg ha^−1^ P_2_O_5_ (calcium superphosphate) and 120 kg ha^−1^ K_2_O (potassium chloride) was applied as base fertilizer in the ridge before planting. The seedlings were planted in 5 rows with 90 cm between rows and 20 cm between hills. The temperature and rainfall of growing location during sweet potato growth stage are presented in Table 6.

### 3.2. Isolation of Starch

Starches from fresh root tubers were prepared following the method of Guo et al. [2]. Briefly, the root tubers were washed cleanly and cut into some pieces. The sample was homogenized in H_2_O with a home blender. The tissue suspension was filtered through four layers of gauze, and the residue was homogenized and filtered again to release more starch. The filtrate was filtered through 100-, 200-, and 300-mesh sieve, successively, and centrifuged (3000× *g*, 5 min). The starch precipitate was washed 5 times with H_2_O and 3 times with anhydrous ethanol. Finally, the starch was dried at 40 °C for 2 d, and ground into powder through 100-mesh sieve.

### 3.3. Granule Morphology and Size Analysis of Starch

Isolated starch in 25% glycerol was viewed and photographed with a polarized light microscope (BX53, Olympus, Tokyo, Japan) under normal and polarized light. Starch size was analyzed with a laser size analyzer (Mastersizer 2000, Malvern, Worcestershire, UK) according to the procedures of Guo et al. [2]. The obscuration of starch–water suspension was about 12%, and the sample was stirred at 2000 rpm during analysis.

### 3.4. Analysis of Starch–Iodine Absorption

Starch was dispersed in dimethyl sulfoxide and stained with iodine solution according to the procedures of Man et al. [24]. Briefly, 10 mg starch in 5 mL dimethyl sulfoxide containing 10% 6.0 M urea was heated at 95 °C for 1 h. The starch suspension was swirled at intervals of 15 min during heating. The dispersed amylose and amylopectin (1 mL) was colorized for 20 min with iodine solution (1 mL of 0.2% I_2_ and 2% KI) in 50 mL volumetric flask diluted with H_2_O. The sample was scanned from 400 to 900 nm using a spectrophotometer (BioMate 3S, Thermo Scientific, Chino, CA, USA).

### 3.5. Measurement of Amylose Content

The apparent amylose content was determined using the absorption value of starch–iodine complex at 620 nm [24]. The true amylose content was measured using concanavalin A precipitation method through an Amylose/Amylopectin Assay Kit (K-AMYL, Megazyme, Bray, Ireland).

### 3.6. XRD Analysis of Starch

Starch was analyzed with an X-ray diffractometer (XRD) (D8, Bruker, Karlsruhe, Germany). Before analysis, the sample was moisturized for 1 week at 25 °C in a desiccator containing a saturated solution of NaCl with a relative humidity about 75%. The testing setting contained an X-ray beam at 200 mA and 40 kV and scanning range of diffraction angle from 2θ 3 to 40° with a step size of 0.02°. The relative crystallinity (RC) was evaluated with the percentage of diffraction peak area to total diffraction area over the diffraction angle 2θ 4 to 30° following the method of Wei et al. [40].

### 3.7. DSC Analysis of Starch

The 5 mg starch and 15 μL water were mixed and sealed hermetically in an aluminum pan. After equilibrating for 2 h at room temperature, the sample was heated from 25 to 130 °C at 10 °C/min using a differential scanning calorimeter (DSC 200-F3, Netzsch, Selb, Germany).

### 3.8. Statistical Analysis

The statistical differences between data from varieties were detected using one-way analysis of variance (ANOVA) by Tukey’s test using the SPSS 19.0 Statistical Software Program. The Pearson correlation analysis and hierarchical cluster analysis were also evaluated using SPSS 19.0 Statistical Software Program. Prior to the analysis, the normal distribution of structural property parameters was assessed using the Shapiro–Wilk test with SPSS 16.0. Only structural property parameters with their significances of normal distributions over 0.05 were used to evaluate the Pearson correlation and hierarchical cluster analysis.

## 4. Conclusions

Starches from 44 sweet potato varieties originating from 15 countries were investigated for their sizes, amylose contents, crystalline structures, and thermal properties. The D[4,3], AAC, TAC, and ΔAC (AAC–TAC) ranged from 8.01 to 15.30 μm, from 19.2% to 29.2%, from 14.2% to 20.2%, and from 4.0% to 11.8% among the 44 starches, respectively. Starches had A-, C_A_-, C_C_-, and C_B_-type with RC from 19.5% to 27.3%. One-, two-, and three-peak DSC curves were detected, and the To, Tp, Tc, and ΔT ranged from 52.2 to 73.8 °C, from 59.7 to 80.0 °C, from 75.3 to 86.9 °C, and from 13.1 to 29.2 °C among the 44 starches, respectively. Based on starch property parameters, the 44 sweet potato varieties were divided into different groups. This research offers references for the utilization of sweet potato germplasm resources.

## Figures and Tables

**Figure 1 molecules-27-01905-f001:**
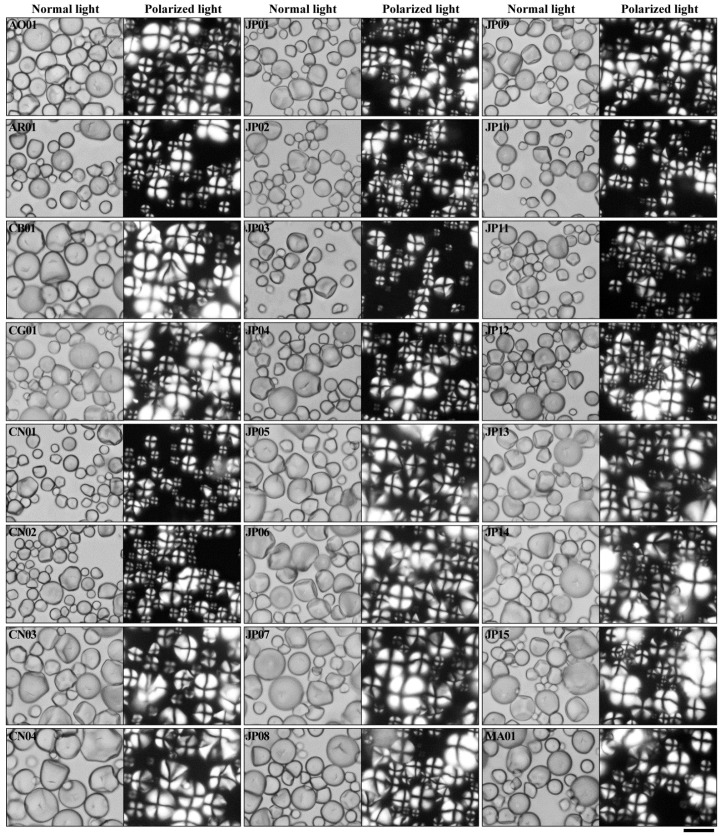
Morphologies of starch granules under normal light and polarized light. Scale bar = 20 μm.

**Figure 2 molecules-27-01905-f002:**
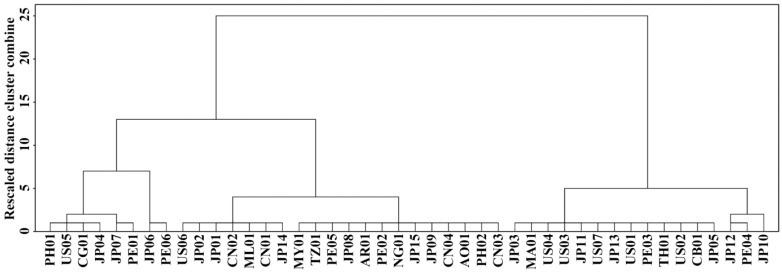
Hierarchical cluster analysis of 44 sweet potato varieties based on OD680, OD620/550, AAC, and TAC in Table 2.

**Figure 3 molecules-27-01905-f003:**
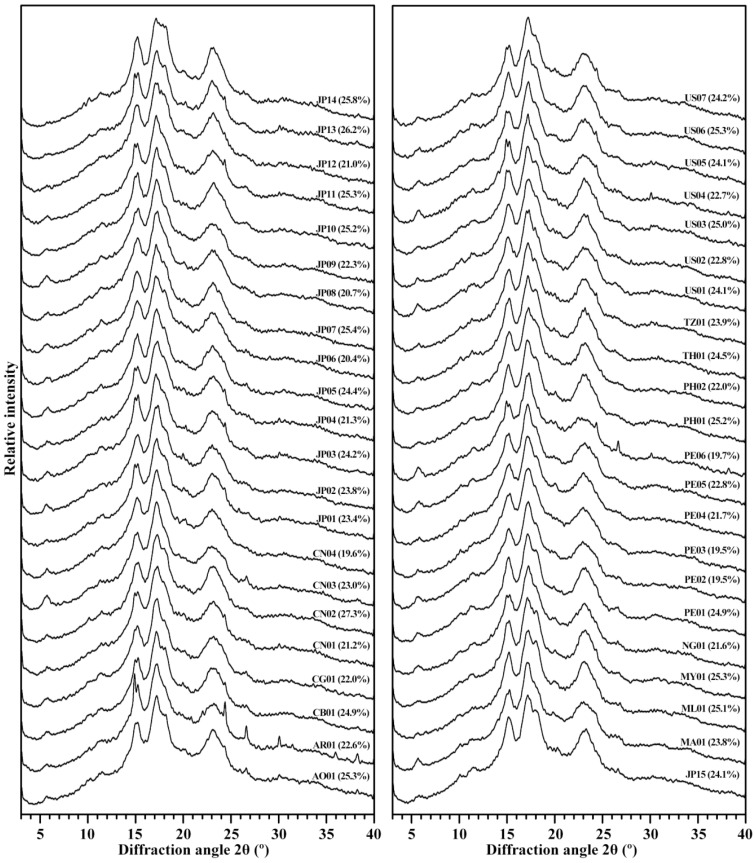
XRD patterns of starches from different sweet potato varieties. The relative crystallinity (RC) is given in the parenthesis after accession ID.

**Figure 4 molecules-27-01905-f004:**
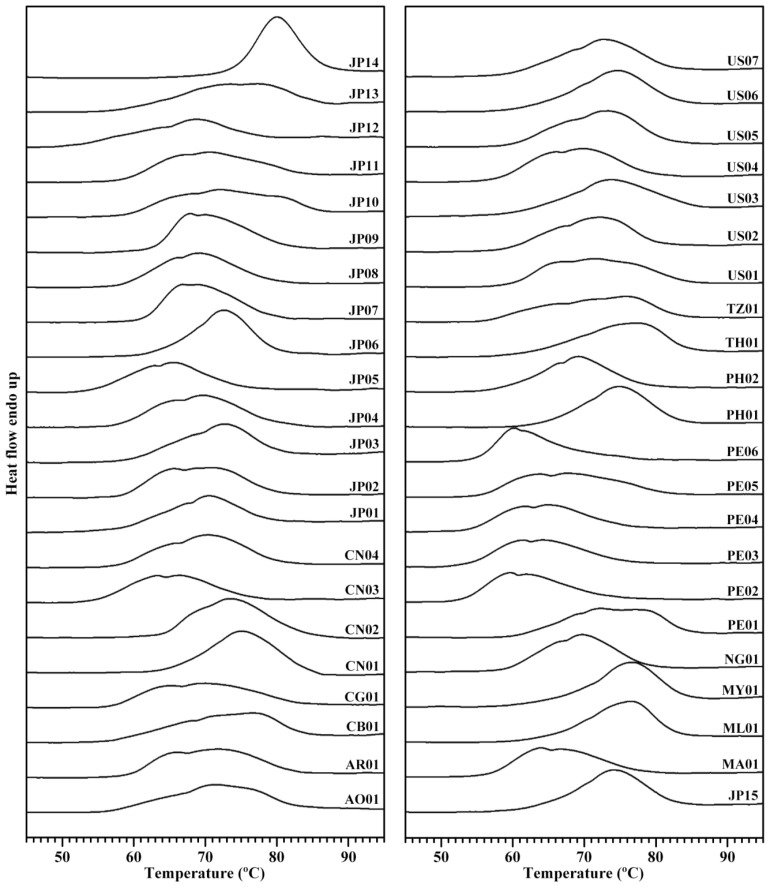
DSC thermograms of starches from different sweet potato varieties.

**Figure 5 molecules-27-01905-f005:**
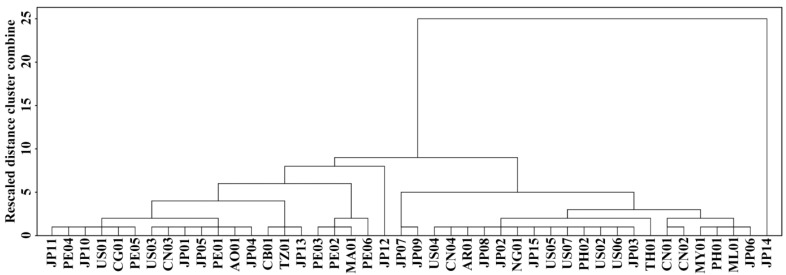
Hierarchical cluster analysis of the 44 sweet potato varieties based on Tp, Tc, ΔT, and ΔH in Table 3.

**Figure 6 molecules-27-01905-f006:**
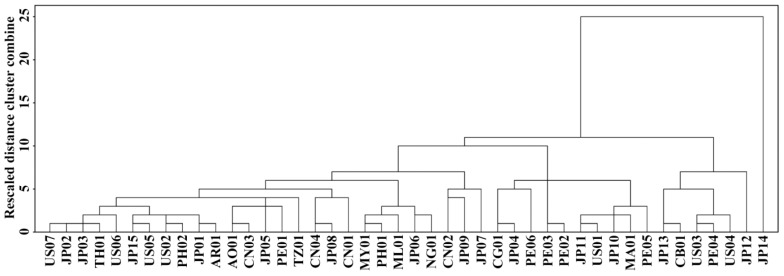
Hierarchical cluster analysis of the 44 sweet potato varieties based on D[4,3], OD680, OD620/550, AAC, TAC, RC, Tp, Tc, ΔT, and ΔH in Table 4.

**Table 1 molecules-27-01905-t001:** Granule size distribution of starches from 44 sweet potato varieties.

Accession ID	d(0.2) (μm)	d(0.5) (μm)	d(0.8) (μm)	D[3,2] (μm)	D[4,3] (μm)
AO01	7.573 ± 0.002 mn	12.817 ± 0.005 tu	19.452 ± 0.005 γ	6.226 ± 0.002 s	13.464 ± 0.004 x
AR01	7.480 ± 0.002 lm	10.881 ± 0.003 l	14.967 ± 0.007 l	5.986 ± 0.002 o	11.023 ± 0.005 l
CB01	7.166 ± 0.002 j	10.331 ± 0.003 i	14.071 ± 0.003 h	5.689 ± 0.002 ij	10.383 ± 0.003 h
CG01	6.466 ± 0.001 f	10.084 ± 0.001 h	14.492 ± 0.002 i	5.335 ± 0.001 f	10.333 ± 0.001 gh
CN01	7.586 ± 0.020 mn	10.911 ± 0.036 l	14.808 ± 0.052 k	5.732 ± 0.010 l	10.941 ± 0.036 k
CN02	7.355 ± 0.003 k	11.003 ± 0.004 m	15.402 ± 0.007 m	5.675 ± 0.003 i	11.161 ± 0.005 m
CN03	6.762 ± 0.004 h	11.099 ± 0.006 n	16.654 ± 0.006 r	5.920 ± 0.001 m	11.648 ± 0.005 o
CN04	11.111 ± 0.002 z	14.669 ± 0.001 β	18.705 ± 0.000 z	7.612 ± 0.001 ζ	14.456 ± 0.001 γ
JP01	7.159 ± 0.001 j	11.018 ± 0.004 m	15.878 ± 0.010 o	6.208 ± 0.003 rs	11.399 ± 0.006 n
JP02	6.785 ± 0.003 h	10.426 ± 0.006 j	14.935 ± 0.009 l	5.698 ± 0.002 jk	10.727 ± 0.006 j
JP03	5.283 ± 0.003 a	8.340 ± 0.005 b	12.391 ± 0.009 e	4.932 ± 0.003 c	8.737 ± 0.007 d
JP04	6.398 ± 0.003 f	9.227 ± 0.003 f	12.460 ± 0.003 e	5.039 ± 0.001 d	9.185 ± 0.002 e
JP05	10.568 ± 0.002 y	14.811 ± 0.003 γ	19.935 ± 0.005 δ	7.681 ± 0.001 η	14.929 ± 0.003 δ
JP06	8.966 ± 0.001 u	13.578 ± 0.002 x	19.356 ± 0.004 γ	6.946 ± 0.001 z	13.997 ± 0.003 z
JP07	7.504 ± 0.001 m	12.269 ± 0.003 r	18.221 ± 0.006 x	6.052 ± 0.001 p	12.746 ± 0.004 t
JP08	10.271 ± 0.003 x	13.945 ± 0.005 z	18.245 ± 0.008 x	7.344 ± 0.003 δ	13.890 ± 0.006 y
JP09	7.187 ± 0.070 j	9.931 ± 0.130 g	13.066 ± 0.167 g	5.718 ± 0.038 kl	9.862 ± 0.120 f
JP10	7.297 ± 0.002 k	10.948 ± 0.003 lm	15.379 ± 0.004 m	5.945 ± 0.002 n	11.150 ± 0.003 m
JP11	6.017 ± 0.005 d	8.651 ± 0.006 d	11.652 ± 0.009 c	4.924 ± 0.004 c	8.623 ± 0.007 c
JP12	8.211 ± 0.004 r	12.410 ± 0.003 s	17.660 ± 0.001 v	6.764 ± 0.001 x	12.790 ± 0.002 t
JP13	5.207 ± 0.021 a	8.012 ± 0.031 a	11.189 ± 0.108 a	3.866 ± 0.011 a	8.013 ± 0.051 a
JP14	8.795 ± 0.004 t	12.749 ± 0.007 t	17.508 ± 0.012 u	6.746 ± 0.002 x	12.922 ± 0.008 u
JP15	6.978 ± 0.004 i	10.518 ± 0.006 k	14.773 ± 0.011 k	5.551 ± 0.003 h	10.691 ± 0.008 ij
MA01	5.855 ± 0.002 c	8.992 ± 0.005 e	12.772 ± 0.010 f	5.083 ± 0.002 e	9.176 ± 0.006 e
ML01	7.974 ± 0.004 p	12.117 ± 0.009 q	17.128 ± 0.015 s	6.210 ± 0.003 rs	12.360 ± 0.010 r
MY01	9.431 ± 0.004 w	13.803 ± 0.007 y	19.183 ± 0.013 β	7.513 ± 0.002 ε	14.113 ± 0.009 α
NG01	7.663 ± 0.005 no	12.449 ± 0.011 s	18.598 ± 0.023 y	6.331 ± 0.004 t	13.037 ± 0.015 v
PE01	9.132 ± 0.252 v	14.378 ± 0.123 α	20.965 ± 0.166 ζ	6.968 ± 0.040 α	14.901 ± 0.016 δ
PE02	9.038 ± 0.001 uv	12.832 ± 0.001 u	17.368 ± 0.002 t	7.002 ± 0.001 β	12.955 ± 0.001 u
PE03	7.392 ± 0.002 kl	11.090 ± 0.003 n	15.682 ± 0.003 n	6.194 ± 0.002 r	11.391 ± 0.003 n
PE04	8.187 ± 0.005 qr	11.730 ± 0.006 p	15.961 ± 0.007 o	6.481 ± 0.003 u	11.857 ± 0.006 q
PE05	6.663 ± 0.010 g	10.426 ± 0.013 j	14.943 ± 0.017 l	5.470 ± 0.004 g	10.643 ± 0.013 i
PE06	5.732 ± 0.002 b	8.491 ± 0.005 c	11.799 ± 0.010 d	4.943 ± 0.003 c	8.606 ± 0.006 c
PH01	8.325 ± 0.006 s	14.328 ± 0.011 α	22.233 ± 0.011 η	6.841 ± 0.002 y	15.296 ± 0.007 ε
PH02	7.930 ± 0.005 p	12.871 ± 0.007 u	19.051 ± 0.008 α	6.594 ± 0.002 w	13.409 ± 0.007 x
TH01	7.754 ± 0.003 o	11.517 ± 0.005 o	16.121 ± 0.007 p	6.313 ± 0.003 t	11.747 ± 0.005 p
TZ01	8.105 ± 0.005 q	13.467 ± 0.008 w	20.448 ± 0.012 ε	6.506 ± 0.004 v	14.211 ± 0.008 β
US01	5.797 ± 0.002 bc	8.460 ± 0.003 c	11.488 ± 0.003 b	4.682 ± 0.002 b	8.417 ± 0.002 b
US02	8.318 ± 0.003 s	12.302 ± 0.004 r	17.192 ± 0.009 s	6.846 ± 0.002 y	12.583 ± 0.006 s
US03	7.479 ± 0.003 lm	12.132 ± 0.008 q	17.856 ± 0.017 w	6.124 ± 0.004 q	12.555 ± 0.011 s
US04	6.576 ± 0.004 g	10.584 ± 0.008 k	15.694 ± 0.014 n	5.707 ± 0.004 jk	11.045 ± 0.009 l
US05	7.362 ± 0.003 k	11.482 ± 0.006 o	16.521 ± 0.008 q	5.929 ± 0.003 mn	11.792 ± 0.006 p
US06	9.025 ± 0.001 u	13.153 ± 0.004 v	18.263 ± 0.010 x	7.169 ± 0.001 γ	13.452 ± 0.006 x
US07	6.203 ± 0.002 e	9.900 ± 0.002 g	14.609 ± 0.002 j	5.556 ± 0.001 h	10.321 ± 0.002 g
Sig.	0.322	0.410	0.671	0.849	0.422

The d(0.2), d(0.5), and d(0.8) are the granule size at which 20%, 50%, and 80% of all the granules by volume are smaller. The D[3,2] and D[4,3] are the surface- and volume-weighted mean diameter, respectively. Sig.: the significance of normal distribution of the 44 samples by Shapiro–Wilk test. Data are means ± standard deviations, *n* = 3. The values with different letters in the same column are significantly different (*p* < 0.05).

**Table 2 molecules-27-01905-t002:** Iodine absorption parameters and amylose contents of starches.

Accession ID	OD680	OD620/550	AAC (%)	TAC (%)	ΔAC (%)
AO01	0.307 ± 0.004 ghijk	1.164 ± 0.026 abcdefg	25.6 ± 0.8 ijk	17.4 ± 0.7 efghi	8.2
AR01	0.324 ± 0.006 klmno	1.222 ± 0.023 gh	26.5 ± 0.7 jklm	17.4 ± 0.2 efghi	9.1
CB01	0.281 ± 0.007 bcd	1.192 ± 0.004 abcdefgh	22.2 ± 0.7 cd	16.4 ± 0.1 cdefgh	5.8
CG01	0.329 ± 0.011 no	1.157 ± 0.003 abcd	28.2 ± 0.9 mno	17.7 ± 0.4 fghi	10.5
CN01	0.300 ± 0.007 efghi	1.155 ± 0.001 abc	24.9 ± 0.6 fghij	17.1 ± 0.2 efghi	7.8
CN02	0.306 ± 0.007 ghijk	1.160 ± 0.012 abcde	25.7 ± 0.8 ijkl	18.2 ± 0.4 hij	7.5
CN03	0.308 ± 0.002 ghijkl	1.208 ± 0.011 cdefgh	21.8 ± 0.2 bcd	14.7 ± 0.3 abc	7.1
CN04	0.295 ± 0.005 defghi	1.186 ± 0.017 abcdefgh	23.6 ± 0.5 defgh	15.7 ± 0.6 abcde	7.9
JP01	0.303 ± 0.004 fghij	1.183 ± 0.004 abcdefgh	24.5 ± 0.6 efghi	17.2 ± 0.4 efghi	7.3
JP02	0.294 ± 0.004 defghi	1.191 ± 0.007 abcdefgh	23.5 ± 0.2 defgh	16.6 ± 0.5 defgh	6.9
JP03	0.291 ± 0.005 cdefgh	1.218 ± 0.019 efgh	23.2 ± 0.7 cdef	17.7 ± 0.3 fghi	5.5
JP04	0.336 ± 0.003 o	1.185 ± 0.011 abcdefgh	28.3 ± 0.4 mno	17.7 ± 0.3 fghi	10.6
JP05	0.299 ± 0.007 defghi	1.237 ± 0.028 h	23.4 ± 0.9 cdefgh	17.1 ± 0.9 efghi	6.3
JP06	0.327 ± 0.007 mno	1.158 ± 0.015 abcd	27.9 ± 0.6 mno	16.1 ± 0.3 bcdefg	11.8
JP07	0.337 ± 0.003 o	1.186 ± 0.001 abcdefgh	28.5 ± 0.2 no	17.2 ± 0.5 efghi	11.3
JP08	0.304 ± 0.001 ghij	1.174 ± 0.012 abcdefg	25.0 ± 0.4 fghij	16.4 ± 0.1 cdefgh	8.6
JP09	0.312 ± 0.007 ijklmn	1.200 ± 0.006 abcdefgh	25.5 ± 0.9 ijk	17.1 ± 0.2 efghi	8.4
JP10	0.304 ± 0.009 ghij	1.167 ± 0.016 abcdefg	25.2 ± 1.1 ghij	20.2 ± 0.9 k	5.0
JP11	0.291 ± 0.006 cdefgh	1.181 ± 0.002 abcdefgh	23.4 ± 0.5 cdefgh	17.6 ± 0.4 efghi	5.8
JP12	0.260 ± 0.005 a	1.209 ± 0.015 cdefgh	19.3 ± 0.5 a	15.0 ± 0.5 abcd	4.3
JP13	0.282 ± 0.003 bcde	1.189 ± 0.025 abcdefgh	22.3 ± 0.7 cd	16.7 ± 0.7 defgh	5.6
JP14	0.326 ± 0.002 mno	1.169 ± 0.019 abcdefg	28.3 ± 0.2 nmo	19.7 ± 0.4 jk	8.6
JP15	0.310 ± 0.008 hijklm	1.161 ± 0.015 abcdef	25.8 ± 0.8 ijkl	17.2 ± 0.3 efghi	8.6
MA01	0.319 ± 0.008 jklmno	1.192 ± 0.014 abcdefgh	22.9 ± 0.8 cde	17.4 ± 0.8 efghi	5.5
ML01	0.303 ± 0.003 fghij	1.160 ± 0.014 abcde	25.4 ± 0.5 hijk	17.4 ± 0.0 efghi	8.0
MY01	0.323 ± 0.008 klmno	1.142 ± 0.006 a	27.8 ± 0.4 mno	18.3 ± 0.6 hij	9.5
NG01	0.325 ± 0.005 lmno	1.179 ± 0.014 abcdefgh	27.4 ± 0.6 lmno	17.8 ± 0.6 fghi	9.6
PE01	0.382 ± 0.013 p	1.146 ± 0.015 ab	29.2 ± 1.3 o	17.7 ± 0.7 fghi	11.5
PE02	0.284 ± 0.008 bcdef	1.197 ± 0.028 abcdefgh	22.1 ± 0.7 cd	14.5 ± 0.3 ab	7.6
PE03	0.281 ± 0.002 bcd	1.206 ± 0.011 cdefgh	21.7 ± 0.3 bcd	16.1 ± 0.6 bcdefg	5.6
PE04	0.259 ± 0.010 a	1.212 ± 0.021 cdefgh	19.2 ± 0.9 a	15.2 ± 0.1 abcd	4.0
PE05	0.296 ± 0.009 defghi	1.158 ± 0.011 abcd	24.4 ± 0.8 efghi	16.0 ± 0.9 bcdef	8.4
PE06	0.303 ± 0.006 fghij	1.160 ± 0.010 abcde	25.0 ± 0.6 fghij	14.2 ± 0.1 a	10.8
PH01	0.331 ± 0.002 o	1.180 ± 0.010 abcdefgh	27.8 ± 0.2 mno	17.7 ± 0.3 fghi	10.1
PH02	0.303 ± 0.007 fghij	1.190 ± 0.003 abcdefgh	24.3 ± 0.5 efghi	16.5 ± 0.7 cdefgh	7.8
TH01	0.299 ± 0.003 defghi	1.193 ± 0.038 abcdefgh	24.3 ± 0.2 efghi	18.0 ± 0.2 ghij	6.3
TZ01	0.320 ± 0.007 jklmno	1.162 ± 0.044 abcdef	27.2 ± 0.5 klmn	17.9 ± 0.1 fghi	9.3
US01	0.298 ± 0.002 defghi	1.177 ± 0.010 abcdefg	24.4 ± 0.2 efghi	18.2 ± 0.4 hij	6.2
US02	0.311 ± 0.003 ijklm	1.205 ± 0.010 bcdefgh	25.3 ± 0.3 ghijk	18.7 ± 0.2 ijk	6.6
US03	0.274 ± 0.008 abc	1.220 ± 0.018 fgh	21.5 ± 1.0 bc	16.2 ± 0.1 bcdefg	5.3
US04	0.270 ± 0.003 ab	1.203 ± 0.018 bcdefgh	20.2 ± 0.3 ab	15.2 ± 0.7 abcd	5.0
US05	0.331 ± 0.007 o	1.178 ± 0.010 abcdefg	27.8 ± 0.9 mno	17.7 ± 0.4 fghi	10.1
US06	0.294 ± 0.003 defghi	1.215 ± 0.026 defgh	23.4 ± 0.1 cdefg	16.6 ± 0.2 defgh	6.8
US07	0.290 ± 0.006 cdefg	1.193 ± 0.039 abcdefgh	23.4 ± 0.8 cdefg	17.5 ± 0.3 efghi	5.9
Sig.	0.094	0.588	0.306	0.171	0.336

AAC: apparent amylose content; TAC: true amylose content; ΔAC: the difference between AAC and TAC (AAC–TAC); Sig.: the significance of normal distribution of the 44 samples by Shapiro–Wilk test. Data are means ± standard deviations, *n* = 3. The values with different letters in the same column are significantly different (*p* < 0.05).

**Table 3 molecules-27-01905-t003:** Thermal property parameters of starches.

Accession ID	To (°C)	Tp (°C)	Tc (°C)	ΔT (°C)	ΔH (J/g)
AO01	57.4 ± 0.6 fgh	71.6 ± 0.1 klm	82.7 ± 0.6 jklmn	25.3 ± 1.2 klmnop	13.6 ± 0.1 bcdefghijk
AR01	59.8 ± 0.6 jklm	71.8 ± 0.4 lmn	82.9 ± 0.0 jklmno	23.2 ± 0.6 ghijklm	14.3 ± 0.0 fghijklm
CB01	56.7 ± 0.5 def	76.5 ± 0.4 vw	84.5 ± 0.3 nopqr	27.9 ± 0.2 qr	14.2 ± 0.3 efghijklm
CG01	57.3 ± 0.3 efg	70.5 ± 0.6 ij	84.2 ± 1.0 mnopqr	26.9 ± 0.7 opqr	13.8 ± 0.0 cdefghijkl
CN01	64.2 ± 0.2 qr	75.3 ± 0.3 tu	85.4 ± 0.4 pqrst	21.3 ± 0.2 cdefgh	16.9 ± 0.5 o
CN02	64.0 ± 0.0 pqr	73.6 ± 0.1 pqr	84.8 ± 0.4 opqr	20.8 ± 0.4 cdefg	16.8 ± 0.5 o
CN03	54.4 ± 0.2 bc	65.8 ± 0.7 d	76.6 ± 1.2 ab	22.2 ± 1.0 efghi	12.6 ± 0.3 bcd
CN04	58.0 ± 0.1 fghi	70.2 ± 0.0 ij	80.4 ± 0.1 defgh	22.5 ± 0.1 efghi	14.7 ± 0.0 ijklm
JP01	58.3 ± 0.1 fghij	70.5 ± 0.1 ij	81.5 ± 0.4 efghij	23.3 ± 0.5 ghijklm	13.0 ± 0.2 bcdefg
JP02	58.7 ± 0.0 ghijk	70.6 ± 0.6 ijk	80.0 ± 0.4 defg	21.3 ± 0.4 cdefgh	14.7 ± 0.2 hijklm
JP03	59.6 ± 0.0 ijkl	72.9 ± 0.1 op	82.4 ± 0.8 ijklm	22.8 ± 0.8 fghijk	13.7 ± 0.1 cdefghijkl
JP04	55.6 ± 1.5 cd	69.9 ± 0.1 hij	81.7 ± 1.7 fghijk	26.2 ± 2.2 nopq	15.0 ± 0.6 jklmn
JP05	54.3 ± 0.1 bc	65.7 ± 0.1 d	75.7 ± 1.1 ab	21.4 ± 1.2 cdefgh	12.3 ± 0.0 bc
JP06	62.4 ± 0.2 no	72.6 ± 0.1 mno	80.3 ± 0.3 defgh	18.0 ± 0.5 b	14.9 ± 0.3 jklmn
JP07	61.0 ± 0.4 lmn	66.8 ± 0.1 e	79.7 ± 0.1 de	18.7 ± 0.3 bc	14.7 ± 0.3 hijklm
JP08	58.1 ± 0.0 fghij	69.1 ± 0.1 gh	80.5 ± 0.5 defghi	22.4 ± 0.5 efghi	14.6 ± 0.1 hijklm
JP09	62.6 ± 0.2 nopq	68.0 ± 0.1 f	82.0 ± 0.0 hijkl	19.5 ± 0.2 bcd	15.3 ± 0.2 mn
JP10	59.1 ± 0.1 hijk	72.1 ± 0.1 mno	86.1 ± 0.1 rst	27.0 ± 0.0 pqr	14.6 ± 0.3 hijklm
JP11	58.8 ± 0.1 ghijk	70.8 ± 0.1 jkl	84.2 ± 0.0 mnopqr	25.4 ± 0.1 lmnopq	14.1 ± 0.6 efghijklm
JP12	52.2 ± 0.3 a	68.8 ± 0.1 fg	80.1 ± 0.1 defgh	27.9 ± 0.1 qr	10.7 ± 0.5 a
JP13	57.5 ± 0.0 fgh	77.1 ± 0.2 wx	86.7 ± 0.0 st	29.2 ± 0.0 r	14.2 ± 0.6 fghijklm
JP14	73.8 ± 0.0 s	80.0 ± 0.0 y	86.9 ± 0.1 t	13.1 ± 0.1 a	16.2 ± 0.1 no
JP15	62.5 ± 0.7 nop	74.1 ± 0.0 rs	83.1 ± 0.1 jklmno	20.6 ± 0.8 cdefg	14.3 ± 0.1 fghijklm
MA01	56.8 ± 0.4 def	63.9 ± 0.1 c	78.8 ± 0.4 cd	22.1 ± 0.1 defghi	12.7 ± 0.0 bcde
ML01	64.1 ± 1.1 pqr	76.7 ± 0.3 vwx	83.6 ± 0.6 klmnop	19.5 ± 1.8 bcd	14.1 ± 0.7 efghijklm
MY01	64.4 ± 1.2 r	76.8 ± 0.0 vwx	84.9 ± 0.1 opqrs	20.5 ± 1.1 cdefg	15.0 ± 0.7 jklmn
NG01	59.1 ± 0.0 hijk	69.9 ± 0.1 hij	79.3 ± 0.2 d	20.2 ± 0.2 bcdef	14.0 ± 0.2 defghijklm
PE01	60.3 ± 0.0 klm	72.5 ± 0.4 mno	84.7 ± 0.6 opqr	24.4 ± 0.6 ijklmno	14.7 ± 0.2 hijklm
PE02	53.3 ± 0.3 ab	59.9 ± 0.2 a	75.3 ± 0.1 a	22.0 ± 0.2 defghi	13.0 ± 0.3 bcdefg
PE03	53.9 ± 0.0 b	61.4 ± 0.1 b	76.5 ± 0.8 ab	22.6 ± 0.8 efghij	12.8 ± 0.2 bcde
PE04	54.6 ± 0.5 bc	65.0 ± 0.0 d	77.4 ± 0.6 bc	22.9 ± 0.1 fghijkl	12.3 ± 0.1 b
PE05	55.8 ± 0.1 cde	67.9 ± 0.3 f	81.8 ± 0.3 fghijk	26.1 ± 0.4 nopq	13.3 ± 0.1 bcdefghi
PE06	54.8 ± 0.3 bc	59.7 ± 0.7 a	77.2 ± 0.2 bc	22.4 ± 0.1 efghi	12.9 ± 0.7 bcdef
PH01	63.8 ± 0.2 opqr	75.0 ± 0.1 st	83.6 ± 0.2 klmnop	19.8 ± 0.0 bcde	14.9 ± 0.5 jklmn
PH02	57.8 ± 0.1 fgh	69.6 ± 0.4 ghi	78.8 ± 0.1 cd	21.0 ± 0.0 cdefg	13.2 ± 0.1 bcdefgh
TH01	61.0 ± 0.2 lmn	77.5 ± 0.0 x	84.9 ± 0.1 opqrs	23.9 ± 0.3 hijklmn	15.1 ± 0.5 klmn
TZ01	56.6 ± 0.1 def	76.0 ± 0.0 uv	84.0 ± 0.4 lmnopq	27.4 ± 0.5 pqr	13.5 ± 0.1 bcdefghij
US01	60.2 ± 0.1 klm	71.8 ± 0.4 lmn	85.3 ± 0.6 pqrst	25.1 ± 0.6 jklmnop	14.9 ± 0.1 jklmn
US02	59.0 ± 1.0 ghijk	71.6 ± 0.7 klmn	80.6 ± 0.1 defghi	21.6 ± 1.1 defgh	13.7 ± 0.3 cdefghijk
US03	60.3 ± 0.1 klm	74.0 ± 0.3 qrs	85.8 ± 0.8 qrst	25.5 ± 0.9 mnopq	14.4 ± 0.3 ghijklm
US04	57.9 ± 0.1 fghi	69.9 ± 0.1 hij	79.9 ± 0.1 def	22.0 ± 0.1 defghi	14.9 ± 0.3 jklmn
US05	60.0 ± 0.1 klm	73.1 ± 0.1 opq	81.9 ± 0.3 ghijk	21.9 ± 0.1 defghi	15.2 ± 0.5 lmn
US06	61.4 ± 0.7 mn	74.5 ± 0.4 rst	83.6 ± 0.1 klmnop	22.2 ± 0.8 defghi	14.3 ± 0.9 fghijklm
US07	59.5 ± 0.0 ijkl	72.7 ± 0.4 nop	82.0 ± 0.3 hijkl	22.5 ± 0.3 efghi	14.2 ± 0.5 efghijklm
Sig.	0.009	0.180	0.189	0.128	0.224

To, Tp and Tc: onset, peak, and conclusion temperature of gelatinization, respectively; ΔT and ΔH: gelatinization temperature range and enthalpy, respectively; Sig.: the significance of normal distribution of the 44 samples by Shapiro–Wilk test. Data are means ± standard deviations, *n* = 3. The values with different letters in the same column are significantly different (*p* < 0.05).

**Table 4 molecules-27-01905-t004:** Correlation coefficients between starch property parameters.

	D[4,3]	OD680	OD620/550	AAC	TAC	RC	Tp	Tc
OD680	0.273							
OD620/550	−0.066	−0.501 **						
AAC	0.251	0.908 **	−0.619 **					
TAC	0.006	0.483 **	−0.304 *	0.593 **				
RC	−0.036	0.164	−0.106	0.187	0.583 **			
Tp	0.122	0.183	−0.300 *	0.343 *	0.602 **	0.625 **		
Tc	−0.144	0.201	−0.416 **	0.354 *	0.646 **	0.632 **	0.858 **	
ΔT	−0.327 *	−0.289	0.070	−0.323 *	−0.113	0.018	−0.015	0.205
ΔH	−0.029	0.372 *	−0.423 **	0.522 **	0.498 **	0.338 *	0.590 **	0.623 **

The analysis is based on property parameters with the significance of normal distribution over 0.05. The RC is relative crystallinity, and its significance of normal distribution is 0.067. The other abbreviations and their normal distributions are listed in Table 1, Table 2 and Table 3. * and ** indicate the significance at the *p* < 0.05 and *p* < 0.01 levels, respectively.

**Table 5 molecules-27-01905-t005:** List of sweet potato accessions used in this study.

Accession ID Used in This Study	Origin Accession ID in China National Sweetpotato Genebank	Accession Name without Origin Accession ID in China National Sweetpotato Genebank	Original Area
AO01		Angola	Angola
AR01	SY00332		Argentina
CB01		Cambodia 1	Cambodia
CG01		Congo 2	Congo
CN01	SY00018		China
CN02	SY00019		China
CN03	SY00192		China
CN04	SY00215		China
JP01	SY00075		Japan
JP02	SY00081		Japan
JP03	SY00083		Japan
JP04	SY00087		Japan
JP05	SY00145		Japan
JP06	SY00147		Japan
JP07	SY00148		Japan
JP08	SY00159		Japan
JP09	SY00369		Japan
JP10		Ayamurasaki	Japan
JP11		Hongyao 2	Japan
JP12		Hongyou	Japan
JP13		Japanese Black	Japan
JP14		Kokei 14	Japan
JP15		Okinawa 100	Japan
MA01	SY00089		Morocco
ML01	SY00091		Mali
MY01	SY00130		Malaysia
NG01	SY00219		Nigeria
PE01	SY00175		Peru
PE02	SY00183		Peru
PE03	SY00190		Peru
PE04	SY00283		Peru
PE05		Peru 1	Peru
PE06		Peru 384	Peru
PH01	SY00114		Philippines
PH02	SY00118		Philippines
TH01		Taizi 1506	Thailand
TZ01		Tanzania	Tanzania
US01	SY00001		United States
US02	SY00003		United States
US03	SY00004		United States
US04	SY00006		United States
US05	SY00014		United States
US06	SY00025		United States

US07	SY00277		United States

**Table 6 molecules-27-01905-t006:** The climatic data of Xuzhou city in 2020.

Month	Average Low Temperature (°C)	Average High Temperature (°C)	Total Rainfall (mm)
Apr	10 ± 4	21 ± 4	16
May	18 ± 3	27 ± 5	36
Jun	22 ± 2	30 ± 4	340
Jul	22 ± 2	29 ± 3	359
Aug	25 ± 2	32 ± 3	72
Sep	19 ± 2	29 ± 3	2
Oct	11 ± 4	20 ± 2	18

## Data Availability

The data are available upon request from the corresponding author.

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
