# Peer review of "Sizes, Components, Crystalline Structure, and Thermal Properties of Starches from Sweet Potato Varieties Originating from Different Countries"

_molecules, 2022, doi:10.3390/molecules27061905_

Round 1

Reviewer 1 Report

The submitted manuscript “Sizes, Components, Crystalline Structure, and Thermal Properties of Starches from Sweet Potato Varieties Originated from Different Countries" is well documented and explained. The manuscript can be accepted for publication after incorporating suggested improvement.

Introduction

  1. “provides foods and” – replace “foods” with “food”
  2. Author has only mentioned the name of scientist (reference 10, 12, 13, 14) who worked on varieties of sweet potato but have not explained their findings and scientific reason behind the differences in starches. Elaborate this part.
  3. Knowledge gap and need of the research is not well defined. Hypothesis of the experiment is not clear. Since different varieties are developed to adopt to certain climatic conditions, explain why it is necessary to varietal difference irrespective of growing conditions.

Materials and Methods

  • Plant Materials: What was the basis for selecting the mentioned countries and varieties of sweet potatoes in the present study?
  • Write about growing conditions like soil type, available soil nutrients, temperature, relative humidity etc.
  • Thermal Property Analysis of Starch: Which thermal properties were analyzed?

Results and Discussion

  • Results are presented well and justification appropriate is provided for obtained results.
  • Granule Size Distribution of Starch: “large size variation of 44 starches were mainly due to their different genotype backgrounds” Explain in detail the difference in genotype that is responsible for difference in granular size.

Tables: It is not clear which variety is significant different from others. The superscript notations like “a, b, c, …” can be used to indicate the difference in all tables.

Reviewer 2 Report

In this paper, the authors present the results of an extensive study listing and comparing selected physicochemical properties of native starch extracted from a large number of different sweet potato cultivars grown in similar conditions. This article is thus mainly a database of all the collected data that is useful to draw some correlations. My main regret is the lack of images to complete this significant set of data. Polarized light optical or scanning electron microscopy images would have been most helpful for morphology comparison. Indeed, the variations in granule average size given in Table 1 is difficult to interpret without knowing the shape of the granules. It is possible that some cultivars produce granules that exhibit a more anisometric shape or a shape distribution. As a consequence, since the granulometry analysis assumes that the particles are spheres, the precision of the size measurement down to the nanometer (!) is not meaningful if an anisometry of shape distribution exists, and should be reduced. If it is not possible to provide images, the authors should at least discuss these aspects.

My second remark is about the cultivation conditions. I perfectly understand why the conditions were fixed to allow for a meaningful comparison. It is logical. However, the selected conditions are not necessarily those used in the country of origin. In this case, how can we know if the physicochemical properties described here would be the same from plants grown in the country of origin? If data exist, they should be mentioned and compared. At least, the authors should comment on this aspect in the manuscript.

Reviewer 3 Report

no line numbering - it makes it very difficult to evaluate the work !!!!! Therefore, I am asking you to look for:

"50-80% starch in its dry" - if we are talking about a raw material, it shouldn't be the term "dry" 

"The granule size distributions exhibited significant differences among 44 sweet potato starches (Table 1). "- please explain it, it does not follow from the results presented in the tables, no statistical evaluation has been carried out 

"... until the supernatant was clear and colorless" - too vague term for laboratory research 

"... centrifuged (3000 g, 5 min) ..." and "... at 2000 rpm ..." - please standardize 

".... The sample (1 mL) was colorized for 20 min with iodine solution (1 mL of 0.2% I2 and 2% KI) in 48 mL of H2O .." !!!!! - STARCH DOES NOT COLOR WITH IODINE Starch forms inclusion complexes with iodine, and when there are at least 6-8 glucose units in the chain, a characteristic blue complex is formed 

Round 2

Reviewer 1 Report

The authors has revised the manuscript and can be accepted for possible publication.